# Serotonin/5-HT1A Signaling in the Neurovascular Unit Regulates Endothelial CLDN5 Expression

**DOI:** 10.3390/ijms22010254

**Published:** 2020-12-29

**Authors:** Kotaro Sugimoto, Naoki Ichikawa-Tomikawa, Keisuke Nishiura, Yasuto Kunii, Yasuteru Sano, Fumitaka Shimizu, Akiyoshi Kakita, Takashi Kanda, Tetsuya Imura, Hideki Chiba

**Affiliations:** 1Department of Basic Pathology, Fukushima Medical University School of Medicine, Fukushima 960-1295, Japan; sugikota@fmu.ac.jp (K.S.); naoichi2004@yahoo.co.jp (N.I.-T.); nishiura.k.09@gmail.com (K.N.); tetsuyai@gmail.com (T.I.); 2Department of Neuropsychiatry, Fukushima Medical University School of Medicine, Fukushima 960-1295, Japan; yasuto.kunii@gmail.com; 3Department of Neurology and Clinical Neuroscience, Yamaguchi University Graduate School of Medicine, Yamaguchi 755-8505, Japan; yasuteru@yamaguchi-u.ac.jp (Y.S.); fshimizu@yamaguchi-u.ac.jp (F.S.); tkanda@yamaguchi-u.ac.jp (T.K.); 4Department of Pathology, Brain Research Institute, Niigata University, Niigata 951-8585, Japan; kakita@bri.niigata-u.ac.jp

**Keywords:** blood-brain barrier, claudin, tight junction, pericyte, endothelial cell, 5-HT1 receptor, PKA, schizophrenia, psychiatric disorder, co-culture

## Abstract

We previously reported that site-selective claudin-5 (CLDN5) breakdown and protein kinase A (PKA) activation are observed in brain microvessels of schizophrenia, but the underlying molecular basis remains unknown. The 5-HT1 receptors decline the intracellular cAMP levels and inactivate the major downstream PKA, and the 5-HT1A receptor is a promising target for schizophrenia. Therefore, we elucidated the involvement of serotonin/5-HT1A signaling in the endothelial CLDN5 expression. We demonstrate, by immunohistochemistry using post-mortem human brain tissue, that the 5-HT1A receptor is expressed in brain microvascular endothelial cells (BMVECs) and mural cells of the normal prefrontal cortex (PFC) gray matter. We also show that PKA is aberrantly activated not only in BMVECs but also in mural cells of the schizophrenic PFC. We subsequently revealed that the endothelial cell–pericyte tube-like structure was formed in a novel two-dimensional co-culture of human primary BMVECs and a human brain-derived pericyte cell line, in both of which the 5-HT1A receptor was expressed. Furthermore, we disclose that the serotonin/5-HT1A signaling enhances endothelial CLDN5 expression in BMVECs under two-dimensional co-culture conditions. Our findings provide novel insights into the physiological and pathological significance of serotonin/5-HT1A signaling in the region-specific regulation of the blood-brain barrier.

## 1. Introduction

The neurovascular unit (NVU) consists of microvascular cells (endothelial cells, pericytes, and smooth muscle cells), glial cells (astrocytes, oligodendroglia, and microglia), neurons, and the extracellular matrix, and contributes to a variety of physiological and pathological processes in the central nervous system (CNS) [1,2,3,4]. Within the NVU, the microvascular endothelial cells and pericytes are primarily involved in maintaining the integrity of the blood-brain barrier (BBB) that separates the CNS from peripheral blood circulation [2,3,5,6,7]. It is noteworthy that the CNS contains the greatest amount of pericytes in the body, with an endothelial cell–pericyte ratio of 1:1 [8,9].

Claudins (CLDNs) are the structural and functional backbone of tight junctions in vertebrate epithelial and endothelial cell sheets [10,11,12,13]. The CLDN family is composed of more than 20 members in mammals and exhibits distinct expression profiles in tissue- and cell-type-specific manners. Among the CLDN family, CLDN5 is by far the most abundantly expressed in brain microvascular endothelial cells (BMVECs), and absolutely required for the development and maintenance of the BBB [14], representing the tightness of the BBB.

We previously found that the brain region-selective breakdown of the CLDN5 protein appears in patients with schizophrenia [15]. In more detail, CLDN5 expression and disappearance in the prefrontal cortex (PFC) of schizophrenic subjects significantly decreased and increased, respectively, compared with those of the normal controls. In addition, such changes were observed in neither the PFC white matter nor the visual cortex (VC) white or gray matter of schizophrenic patients. Almost coincidently, Greene et al. reported discontinuous expression of CLDN5 in the parietal lobe of schizophrenic patients compared with age-matched normal brains [16]. They also demonstrated that the site-specific suppression of CLDN5 resulted in localized BBB disruption and the onset of schizophrenia-like phenotypes in mice. Furthermore, they revealed that anti-psychotic drugs dose-dependently induced CLDN5 expression. A decreased expression of CLDN5 protein was also detected in the hippocampus gray matter of schizophrenic subjects, but in neither the hippocampus white matter nor the orbitofrontal gray or white matter [17]. Taken collectively, these results highlight the pathobiological relevance of the region-selective CLDN5 breakdown in schizophrenia. However, it remains poorly defined how the expression of CLDN5 protein is diminished in a brain site-specific manner.

We formerly identified that cAMP phosphorylates CLDN5 at Thr207 in a protein kinase A (PKA)-dependent fashion, leading to size-selective loosening of the endothelial barrier against small molecules, despite cAMP inducing the *CLDN5* expression in a PKA-independent manner [18,19]. Interestingly, we subsequently reported that microvascular and perivascular PKA activation appeared to be observed in the schizophrenic PFC but in neither the schizophrenic VC nor the control PFC or VC [15]. More importantly, the phosphorylated PKA (pPKA)-positive BMVECs in the schizophrenic PFC occasionally exhibited focal loss of CLDN5. Taken together with reports showing that cAMP signaling is aberrantly activated in the schizophrenic PFC [20,21], we assumed that the regional cAMP/PKA-related signaling in brain microvessels could participate in the regulation of endothelial CLDN5 expression in normal and schizophrenic brains.

Serotonin (5-hydroxytriptamine; 5-HT)-actuated nerve endings are prominently close to microvessels in PFC, suggesting that the microvascular endothelial cells and mural cells (microvascular pericytes and smooth muscle cells) receive this chemical transmitter from activated neurons [22,23]. Among members of the 5-HT receptors, the 5-HT1 receptor is liganded by serotonin with high affinity and is known to decrease intracellular cAMP levels [24,25]. Moreover, serotonergic signaling is altered in several psychiatric disorders, including schizophrenia, and the 5-HT1A receptor is a promising target for schizophrenia [22,26]. Along this line, in the present study, we focused on the 5-HT1A expression in brain microvascular cells and its functional significance in endothelial CLDN5 expression.

Here, we report that the 5-HT1A receptor is strongly expressed in the microvascular endothelial and mural cells of normal PFC gray matter. We also show that PKA is aberrantly activated not only in microvascular endothelial cells but also in mural cells of the schizophrenic PFC. Moreover, we demonstrate that serotonin/5-HT1 signaling promotes endothelial CLDN5 expression in microvascular endothelial cell–pericyte tubes under two-dimensional co-culture conditions.

## 2. Results

### 2.1. 5-HT1A is Expressed in Microvascular Endothelial Cells and Mural Cells of Normal Human PFC

We first determined, by immunofluorescent staining using post-mortem normal human brain tissue of the PFC gray matter, whether microvascular endothelial and mural cells expressed the 5-HT1A receptor. To this end, we utilized the vascular endothelial marker CD31, the mural cell markers platelet-derived growth factor receptor β (PDGFRβ) and α smooth muscle actin (αSMA), and the neuronal markers class III β-tubulin or microtubule-associated protein 2 (MAP2). As shown in Figure 1 and Appendix A, 5-HT1A was partially colocalized with CD31 and was also observed in the surrounding perivascular cells. Unexpectedly, the signal intensity of 5-HT1A in the brain microvascular cells was higher than that in the brain parenchyma. In addition, 5-HT1A was observed in microvascular smooth muscle cells (PDGFRβ^+^/αSMA^+^) and pericytes (PDGFRβ^+^/αSMA^-^) (Figure 2). Thus, 5-HT1A appeared to be expressed in brain microvascular endothelial cells and mural cells.

### 2.2. PKA is Activated in Microvascular Endothelial Cells and Mural Cells of the Schizophrenic PFC

We subsequently verified which types of microvascular cells in the schizophrenic PFC showed aberrant PKA activation. A strong pPKA signal was detected in the PDGFRβ-expressing mural cells (Figure 3A,B). In addition, pPKA was also colocalized with the endothelial marker CD34. Moreover, a weak pPKA signal was at least partially colocalized with glial fibrillary acidic protein (GFAP)-expressing glia.

### 2.3. 5-HT1A is Expressed in Human Brain Microvascular Endothelial Cell–Pericyte Tubes In Vitro

To gain some mechanistic insights, we then performed a two-dimensional co-culture of human primary BMVECs and the human brain-derived pericyte cell line (HBPCT) [27] (Figure 4A). Surprisingly, the tube-like formation was started and completed one and two days after co-culture, respectively, and maintained at least for an additional two to four days in this two-dimensional co-culture model (Figure 4B). CLDN5 and PDGFRβ were observed along the cultured tubes (Figure 4C and Figure 5). Moreover, the 5-HT1A receptor was detected in the endothelial cell–pericyte tube-like structure in vitro (Figure 5).

### 2.4. Serotonin/5-HT1A Signaling Enhances CLDN5-Immunoreactive Area in Microvascular Endothelial Tube Under Two-Dimensional Co-Culture Conditions

Using the novel two-dimensional co-culture system, we subsequently validated whether serotonin regulated endothelial CLDN5 expression. In the vehicle-treated co-culture, the CLDN5-positive signal was focally diminished, whereas another tight-junction marker ZO-1 displayed a linear expression pattern (Figure 6A). Focal loss of CLDN5 was prevented by the treatment of co-culture with serotonin. The 5-HT1A antagonist WAY reversed the effect of serotonin on endothelial CLDN5 expression. The significant changes in the CLDN5-covered area were shown by quantitative analysis (Figure 6B). Hence, the serotonin/5-HT1A signal in brain microvascular endothelial cells and pericytes appeared to induce endothelial CLDN5 expression.

## 3. Discussion

The receptors 5-HT1A and 5-HT1B are expressed in postsynaptic and presynaptic neurons of the PFC, respectively [28,29,30]. In the present study, we showed, by immunofluorescent staining using post-mortem human brain tissue, that the 5-HT1A receptor was also expressed in microvascular endothelial and mural cells of the normal PFC gray matter. Unpredictably, the immunoreactive signal intensity of 5-HT1A in the brain microvascular cells was stronger rather than that in the brain parenchyma. Analysis of a larger number of cases would be required to draw more solid conclusions about an abundance of 5-HT1A in normal brain microvascular cells. Concerning the expression of 5-HT1 members in pericytes, it should be noted that the expression of 5-HT1B receptors is induced in rat pericytes after a spinal cord injury [31].

We previously found that PKA was activated in microvascular endothelial cells and perivascular cells of schizophrenic PFC, compared with the control PFC [15]. Aberrant PKA activation was not observed in those of the schizophrenic VC, indicating the region-specific alteration in cAMP/PKA signaling in brain microvessels. In the present work, we indeed demonstrated that PKA was abnormally activated not only in brain microvascular endothelial cells but also in mural cells of the schizophrenic PFC. Taken collectively with our previous results showing that cAMP/PKA-dependent phosphorylation of CLDN5 at Thr207 causes size-selective loosening of the endothelial barrier against small molecules [18,19], the site-specific PKA activation in BMVECs and mural cells is most probably responsible for the localized breakdown of CLDN5 and the BBB.

It has long been considered that vessel tubes are not formed under a two-dimensional co-culture [32]. We developed a novel co-culture model of human primary BMVECs and the human brain-derived pericyte cell line (HBPCT). Unexpectedly, however, in this two-dimensional co-culture model, the endothelial cell–pericyte tube-like structures were able to be formed and maintained at least for four days without using special gels or devices. We also confirmed the expression of the 5-HT1A receptor in both brain microvascular endothelial cells and pericytes under the two-dimensional co-culture model.

Another conclusion of our study is that exposure of the endothelial cell–pericyte culture to serotonin induces endothelial CLDN5 expression via the 5-HT1A receptor. This conclusion was drawn from the treatment of the above-mentioned two-dimensional co-culture system with serotonin and the 5-HT1A antagonist WAY. Although WAY also acts as a dopamine D4 receptor (DRD4) agonist [33], DRD4 is expressed in neither human primary BMVECs nor HBPCT (our unpublished results), indicating that the effects of WAY in these cells are mediated through the 5-HT1A. We speculate that serotonin/5-HT1A signaling inhibits the cAMP/PKA pathway in microvascular endothelial cells and mural cells, preventing CLDN5 phosphorylation and subsequent breakdown, from the following findings: (1) the serotonin/5-HT1A decreases intracellular cAMP levels [24,25]; (2) the 5-HT1A receptor is strongly expressed in brain microvascular endothelial and mural cells both in vitro and in vivo (the current study); and (3) the PKA activity and the amount of endothelial CLDN5 are conversely regulated in microvessels [15,18,19]. We also assume that control and serotonin-exposed microvascular cells in vitro (Figure 6A) correspond to schizophrenic and healthy BMVECs in vivo. Because apically-applied serotonin increases the permeability of endothelial monolayers [34], the highest coverage of pericytes in brain microvessels should be essential for the positive regulation of BBB by serotonin/5-HT1A signaling. Furthermore, since gap junctions between endothelial cells promote the barrier function of tight junctions [35], gap-junction channels and/or hemichannels between endothelial cells and mural cells [36,37,38,39] may also contribute to the maintenance of endothelial CLDN5 expression. In fact, cAMP can pass through gap-junction channels and hemichannels. It should also be noted that activated PKA phosphorylates several connexins such as connexin 43, enhancing gap-junction assembly and intercellular communication [40,41,42].

In summary, we have demonstrated that the 5-HT1A receptor is highly expressed in normal human BMVECs and pericytes. We also uncovered that the serotonin/5-HT1A signaling up-regulates endothelial CLDN5 expression in the novel co-culture system. Since 5-HT1A functions as a target for schizophrenia [22,26,43], further studies are required to determine whether 5-HT1A-targeting drugs improve the localized PKA activation and endothelial CLDN5 loss in schizophrenia in future experiments. It would be also interesting to clarify the involvement of serotonin/5-HT1A/cAMP/PKA signaling and gap-junction channels and/or hemichannels not only in schizophrenia but also in other psychiatric disorders.

## 4. Materials and Methods

### 4.1. Antibodies

The antibodies used in the current study are listed in Appendix A. The specificity of the 5-HT1A antibody (N3C1, GeneTex, CA, USA) [44,45] was verified by Western blotting using multiple positive control samples (unpublished data).

### 4.2. Cases and Brain Tissues

Post-mortem human brain tissues were obtained from the Fukushima Postmortem Brain and DNA Bank for Psychiatric Research (Fukushima PMB/DNA Bank). The schizophrenic donors were a 59-year-old female, a 57-year-old male, and a 74-year-old female, and control donors were a 57-year-old female, a 70-year-old male, and an 81-year-old female. They had no history of alcoholism or drug abuse. The brains were collected and cut coronally in 10-mm slabs, from which the PFC (Brodmann area 10) was dissected. The samples were frozen in optimal cutting temperature (OCT) compound (45833, Sakura, Tokyo, Japan) in liquid nitrogen, and stored at −80 °C until use.

### 4.3. Cell Culture

Human primary BMVECs were obtained from Cell Systems and cultured in a Complete Serum-Free Medium Kit With RocketFuel™ (SF-4Z0-500, Cell Systems, WA, USA). The HBPCT was established as described previously [27] and maintained in Dulbecco’s Modified Eagle Medium (DMEM) high glucose with 10% fetal bovine serum at 33 °C These cell lines were passaged every 7 days at 1 to 5 dilution using 0.25% trypsin-EDTA (209-16941, Wako, Tokyo, Japan). For tube formation, 1 × 10^4^ HBPCTs were passaged on a type I collagen (637-00653, Nitta gelatin, Osaka, Japan)-coated glass base dish (3910-035, Iwaki, Tokyo, Japan) and incubated at 37 °C. After 3 d, 5 × 10^3^ BMVECs were added to it and grown in a Complete Serum-Free Medium Kit With RocketFuel™ (SF-4Z0-500, Cell Systems, WA, USA) with vehicle, 50 µM serotonin hydrochloride (H9523, Sigma-Aldrich, MO, USA) and 10 µM 5-HT1A antagonist WAY-100635 maleate (ab120550, abcam, Cambridge, UK), for an additional 4 days, being subjected to immunostaining.

### 4.4. Immunostaining and Imaging

The human brain tissues were sectioned with a 20-μm thickness from snap-frozen tissues and fixed in 100% methanol for 10 min at −20 °C. After blocking with 5% skimmed milk in phosphate buffered saline (PBS) for 20 min, they were incubated overnight at 4 °C with primary antibodies in 2% bovine serum albumin and PBS. They were subsequently reacted with fluorescently-conjugated secondary antibodies after being rinsed three times.

Cultured cells were fixed with 4% paraformaldehyde for 10 min followed by permeabilization with 0.1% Triton-X in PBS. After blocking with 5% skimmed milk in PBS for 30 min, they were stained in the same manner as above.

All samples were examined and photographed using a scanning confocal laser microscopy (FV1000, Olympus, Tokyo, Japan) and processed by ImageJ software (version 1.49, NIH, MD, USA). In immunohistological analyses, similar results were obtained from three control subjects and three schizophrenic ones, and representative images are shown in Figure 1, Figure 2 and Figure 3.

### 4.5. Quantitative Morphometric Analysis and Statistics

For quantification of the CLDN5-covered area in Figure 6, the length of the linearly distributed CLDN5 signal was manually measured using the straight-line tool in ImageJ software (version 1.49, NIH, MD, USA) and subsequently divided by the corresponding length of the ZO-1 signal. A total of six fields from three different wells in each group, of which the file names were blind to the examiner, were measured. The statistical significance of differences was evaluated by the Mann–Whitney U test using SPSS Statistics (version 26, IBM, NY, USA). *p*-values less than 0.05 were considered to indicate a statistically significant result.

## Figures and Tables

**Figure 1 ijms-22-00254-f001:**
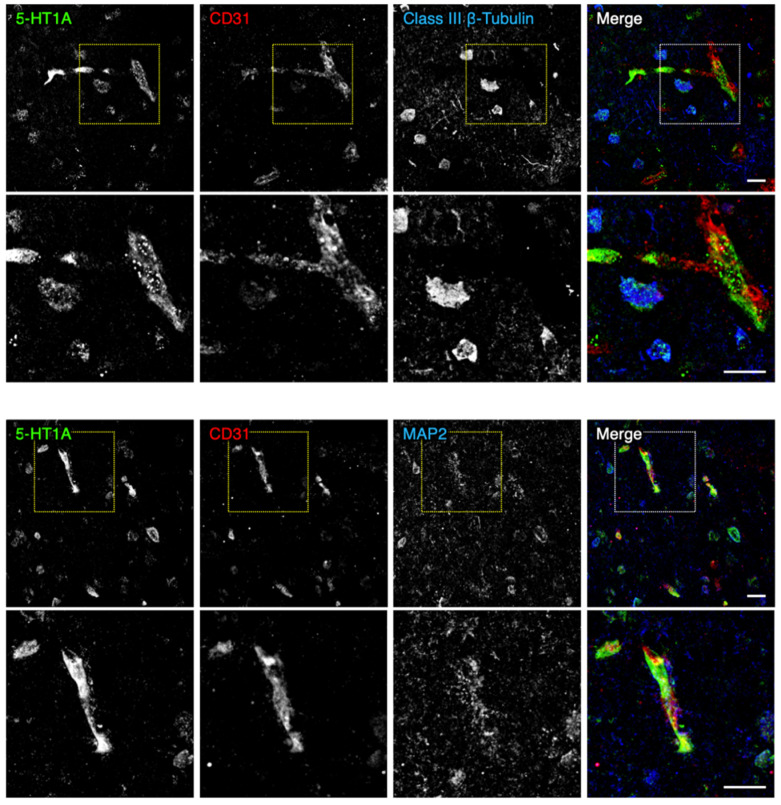
Expression of the 5-HT1A receptor in the microvascular endothelial and perivascular cells of the normal human prefrontal cortex (PFC). Confocal images of the normal PFC gray matter stained for 5-HT1A and CD31 together with either class III β-tubulin or microtubule-associated protein 2 (MAP2). Scale bars, 50 µm.

**Figure 2 ijms-22-00254-f002:**
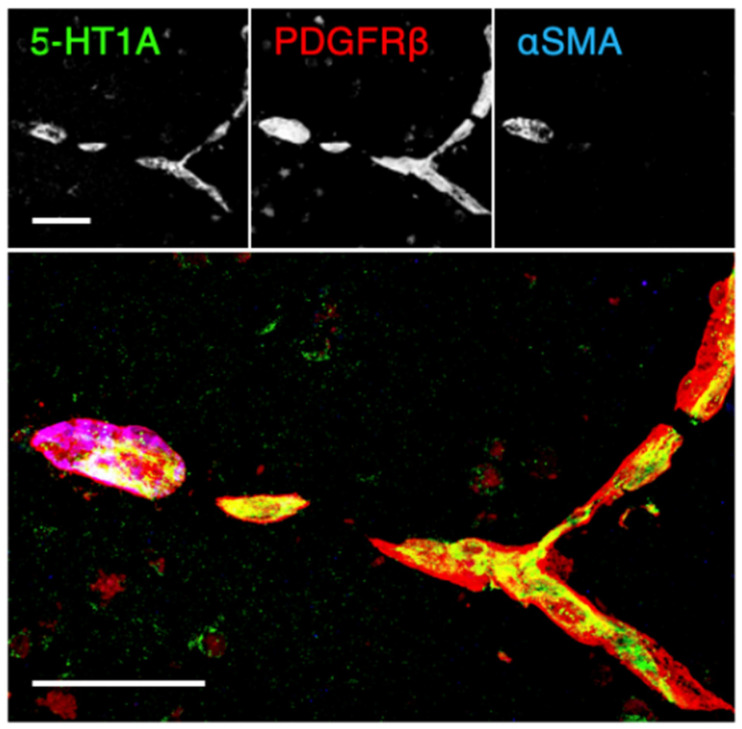
Expression of the 5-HT1A receptor in microvascular pericytes and smooth muscle cells of the normal human PFC. Confocal images of the normal PFC gray matter stained for 5-HT1A, platelet-derived growth factor receptor β (PDGFRβ) and α smooth muscle actin (αSMA). Scale bars, 100 µm.

**Figure 3 ijms-22-00254-f003:**
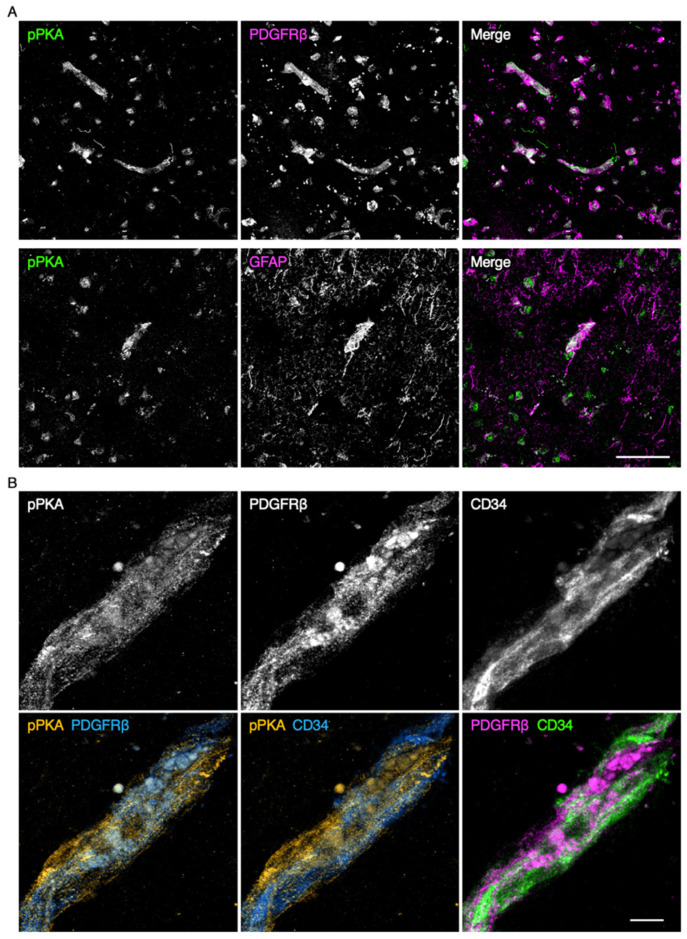
Protein kinase A (PKA) activation in the microvascular endothelial cells and pericytes of the schizophrenic PFC. (**A**) Confocal images of the schizophrenic PFC gray matter stained for phospho-PKA (pPKA) and either PDGFRβ or glial fibrillary acidic protein (GFAP). (**B**) Confocal images of the schizophrenic PFC gray matter stained for pPKA, PDGFRβ, and CD34. Scale bars, 50 µm (upper); 10 µm (lower).

**Figure 4 ijms-22-00254-f004:**
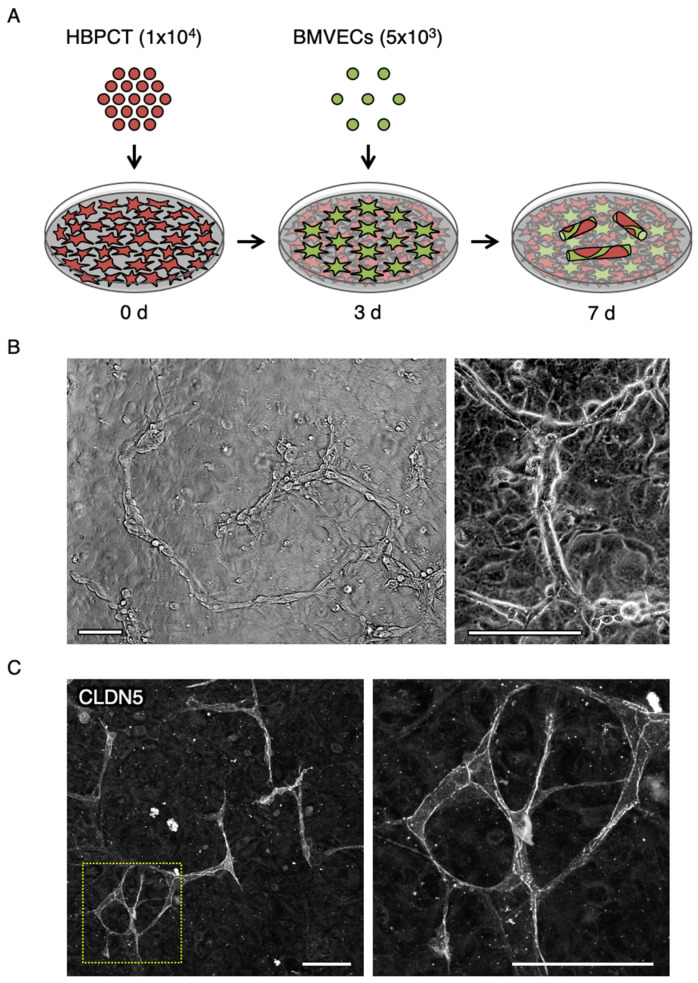
Formation of brain microvascular endothelial cell–pericyte tubes in two-dimensional co-culture. (**A**) Schematic method for two-dimensional co-culture system. HBPCT: human brain-derived pericyte cell line; BMVECs: human primary brain microvascular endothelial cells. (**B**) Phase-contrast micrograph showing tube-like structure. (**C**) Confocal images of two-dimensional co-culture stained for CLDN5. Scale bars, 100 µm.

**Figure 5 ijms-22-00254-f005:**
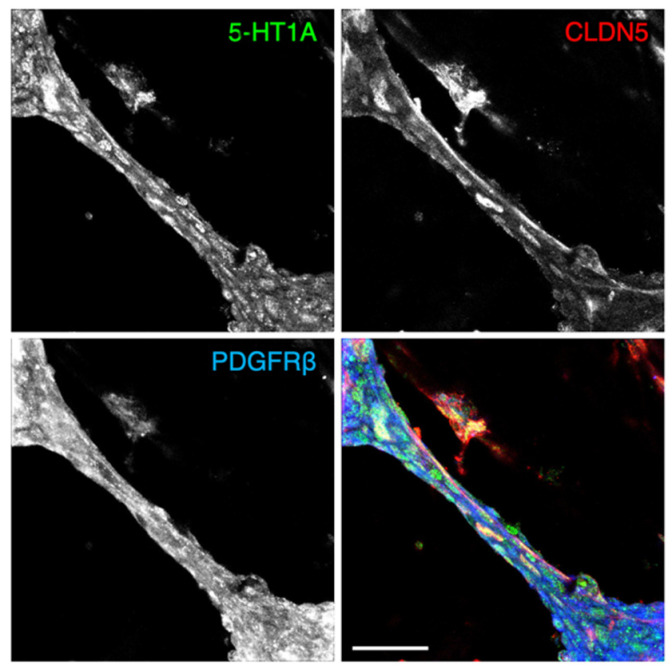
Expression of the 5-HT1A receptor in the human brain microvascular endothelial cell–pericyte tube-like structure in vitro. Confocal images of two-dimensional co-culture stained for HT1A, CLDN5, and PDGFRβ. Scale bar, 50 µm.

**Figure 6 ijms-22-00254-f006:**
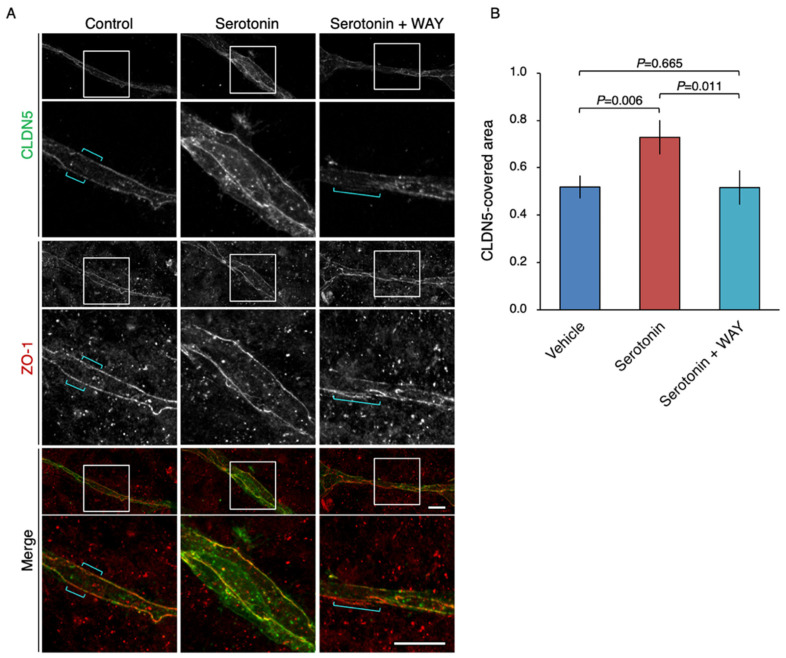
Up-regulation of the CLDN5-immunoreactive area in the microvascular endothelial tube-like structure via the serotonin/5-HT1A receptor signaling. (**A**) Confocal images of two-dimensional co-culture stained for CLDN5 and ZO-1. The HBPCT and the human primary BMVECs were grown under two-dimensional co-culture conditions for 4 days. Brackets indicate the breakdown of CLDN5. WAY: WAY-100635. Scale bars, 50 µm. (**B**) The CLDN5-length is divided by the ZO-1-length, and the relative CLDN5-covered area is shown in histograms (mean ± SD; *n* = 3). Similar results were obtained from three independent experiments.

## Data Availability

All the data used in this study are available from the corresponding author upon request.

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
