# Peer review of "Serotonin/5-HT1A Signaling in the Neurovascular Unit Regulates Endothelial CLDN5 Expression"

_ijms, 2020, doi:10.3390/ijms22010254_

Round 1

Reviewer 1 Report

This is a well-written, concise manuscript that builds on previous work by the authors that showed a brain region–selective decrease in CLDN5 protein, a protein that is critical for blood-brain barrier (BBB) integrity, and upregulation of vascular PKA activity in post-mortem samples from individuals with schizophrenia (ref. 15). This aligns with other reports that the integrity of the  BBB is weakened in schizophrenia, and suggests that this may be due, in part, to a PKA-dependent reduction in CLDN5.

This manuscript purports that: 1) serotonin 5-HT1A receptors are expressed in the microvascular endothelial and mural cells of normal prefrontal cortex (PFC) gray matter; 2) PKA is abnormally activated in microvascular endothelial cells and in mural cells of PFC samples from individuals with schizophrenia; and that 3) 5-HT1A receptor activation promotes endothelial CLDN5 expression in microvascular endothelial cell–pericyte tubes in vitro (likely via its ability to decrease PKA activity).

This is an interesting set of observations with important relevance to schizophrenia pathology, however, there are major issues to address experimentally before firm conclusions can be made about the data:

1) The 5-HT1A antibody used must be validated; there are no negative or positive controls provided showing that the antibody selectivity labels 5-HT1A receptors. Antibodies that label GPCRs are notoriously non-selective (https://www.nature.com/news/reproducibility-crisis-blame-it-on-the-antibodies-1.17586).

2) Provide references or support that phosphorylation (PO4) of CLDN5 leads to its breakdown; ref. 18 notes that PO4 of CLDN5 induces expression of its gene.

3) Lacking in Figure 3 are results from healthy human brain controls.

4) It is concluded from the images in Figure 4 that the endothelial cell–pericyte culture formed "tubes," however there is no confirmation that these are actual tubular structures.

5) There is a logical problem that needs to be addressed. The authors state that "microvascular and perivascular PKA activation appeared to be observed in the schizophrenic PFC but in neither the schizophrenic VC nor the control PFC or VC." These observations suggest that PKA is typically not present (or its activity not detectable) in healthy, normal microvascular and perivascular. How, then, can serotonin activate 5-HT1A reducing PKA activity, to then enhance CLDN5 expression in cultured HBPCT and BMVEC cells, if they typically don't express activated PKA? Please clarify.

Minor issues:

1) Make note that WAY100635 is a selective 5-HT1A antagonist, not a pan 5-HT1 antagonist.

2) It would be helpful to the reader to see zoomed-out images of the samples used for analyses, and similarly, to include the magnification(s) used for taking the images. 

3) Also helpful would be images stained with cresyl violet or another stain to orient the reader to the intact brain structure(s) that were analyzed.

Reviewer 2 Report

In this manuscript, authors present results about the role of 5-HT1A receptor ans serotonin signaling in the neurovascular unit enhancing the endothelial CLDN5 expression. Which, represent a logical extension of previous work. Although the topic is intriguing, there is some points to be clarified. Please, find my comments and suggestions for improving manuscript quality below.

  • Row 85-86. It is presented in the Introduction that “the 5-HT1 receptor shows high affinity for serotonin”. However, the affinity is related to the degree to which the neurotransmitter tends to bind to the receptor. Please, rephrase that.
  • It is not clear in the introduction why the 5-HT1A receptor and only this receptor was chosen as the target of the study.
  • In Figure 1, it is difficult to distinguish the β-tubulin III and MAP2 staining from the background. Please, provide better representative images for these neuronal markers.
  • Row 127: The authors claimed that “pPKA signal was much weaker than that in the microvascular cells”. How was it evaluated?
  • Figure 3A and 3B were indicated in the text and figure caption. However, there is not letter in Figure 3. Please, make it consistent.
  • In the Caption of Figure 6, it was described that the number of samples for quantification of the CLDN5-covered area is = 6. However, in the item 4.5 the number of sample is six fields from three different wells. How many samples per group was used for statistical analysis? Please, clarify that.
  • WAY-100635 is a selective 5-HT1A receptor antagonist. But, it can also act as a dopamine D4 receptor agonist (Chemel et al., 2006; doi: 10.1007/s00213-006-0490-4). Based on that, it is possible that the treatment with WAY alone could interfere with the endothelial CLDN5 expression. It would be interesting to see that result and add that information in the manuscript.
  • Row 176: The authors claimed that “the signal intensity of 5-HT1A in the brain microvascular cells was stronger rather than that in the brain parenchyma”. How was it evaluated?
  • Why the authors choose to analysis the data of Figure 6B using Student's t-test? Are the data normally distributed? Which normality test was used? The appropriate statistical analysis in this case is the one-way analysis of variance.

Round 2

Reviewer 2 Report

The revised manuscript is now considerably improved compare to its initial version. It will be suitable for publication after addressing one minor point:

- Figure 6 caption, the number of samples should be corrected. According to the description “Total six fields from three different wells in each group” (row 267-26), the sample size is 3 instead of 6.

Author Response

Dear the reviewer,

Thank you for your additional comment.

We corrected the sample size in Figure 6 caption from 6 to 3.

Best regards,

Hideki
